# Can ‘Script Elicitation’ Methods Be Used to Promote Physical Activity? An Acceptability Study

**DOI:** 10.3390/bs14070572

**Published:** 2024-07-05

**Authors:** William Peer, Ruth R. Mathews, Xueli Ng, Winson Ho Chun Wong, Benjamin Gardner

**Affiliations:** 1Department of Psychology, University of Surrey, Elizabeth Fry Building, Guildford GU2 7XH, UKwinsonwonghochun@gmail.com (W.H.C.W.); 2Habit Application and Theory (HabitAT) Group, Department of Psychology, University of Surrey, Elizabeth Fry Building, Guildford GU2 7XH, UK

**Keywords:** behavioural science, health behaviour, health promotion, psychology

## Abstract

Sustaining physical activity may require incorporating activity into everyday routines. Yet, many such routines are executed habitually, so people may not recognise physical activity opportunities. ‘Script Elicitation’—a novel intervention method whereby participants detail the content and structure of their routines and are supported to plan modifications to those routines—has not yet been applied to physical activity. This mixed-methods study assessed the acceptability of Script Elicitation for increasing physical activity among office workers. Eleven UK office workers completed the one-to-one Script Elicitation procedure, describing their typical before-, during-, or after-work routines and receiving guidance on incorporating activity into those routines. One week later, they rated the acceptability of the method and completed a semi-structured interview. Physical activity was self-reported at baseline and at the one-week follow-up. Acceptability was descriptively assessed on two quantitative criteria (no clear decrease in physical activity; above-midpoint acceptability scores) and qualitatively explored via Thematic Analysis. The acceptability criteria were met, and participants reported raised awareness of routines and physical activity increases. Script Elicitation appears potentially suitable for promoting activity within everyday routines. If effectiveness is shown in a more rigorous trial, future work will need to develop script-based methods for efficient delivery at scale as a public health intervention.

## 1. Introduction

Regular physical activity (PA) improves health and wellbeing and may extend life by up to 4 years [1,2]. UK guidelines recommend 150 min of moderate PA, or 75 min of vigorous PA, per week [3]. While many fail to meet this target—for example, ~25% of UK adults fail to attain even 30 min/week of moderate PA [4]—a dose–response relationship between PA and health suggests that even those who meet the guidelines could benefit by further increasing PA [5].

PA promotion interventions often have limited long-term impact because people lose motivation over time [6]. PA often competes with other, higher-priority demands; many people report simply being ‘too busy’ to increase their activity, or to maintain PA gains [7]. This has led to calls for interventions to embed PA into stable everyday patterns of behaviour (i.e., routines) [8], rather than promoting PA in a way that obstructs the pursuit of competing goals [9]. Incorporating PA into personalised routines requires an understanding of the content of such routines, to locate opportune moments to be physically active.

Many everyday routines are performed on ‘autopilot’, with little conscious awareness [10]. This may occur for multiple reasons. One is that people mentally represent their actions according to their intended purpose, rather than according to whether those actions involve PA or afford PA opportunities [9,11]. For example, somebody sitting at a bus stop is more likely to see themselves as ‘waiting for the bus’—i.e., an action directed towards the goal of catching a bus—than ‘sitting at a bus stop’ or ‘not moving’ [12]. Additionally, people tend not to pay concurrent attention to the mechanistic ‘sub-actions’ that comprise the action they perceive themselves to be enacting. Somebody who thinks of their action as ‘waiting for the bus’ is unlikely to think about lower-level ‘sub-actions’ involved in ‘waiting for the bus’, such as ‘sitting on a bench’, ‘looking left’, ‘looking at wristwatch’, and so on. In sum, people may fail to recognise their PA—or lack thereof—when engaging in routines that incidentally incur activity. People also lack awareness of routines that are habitual. Habitually executed routines are sequences of sub-actions that proceed automatically, with completion of one sub-action automatically triggering the initiation of the next, until the higher-order action is completed [13,14]. ‘Going for a run’, for example, may involve automated transitions between ‘leaving the house’, ‘locking the door’, ‘walking to the park’, and so on. Habitually executed routines, by nature, require little oversight [13], so people often think about more meaningful, unrelated matters when performing them [10]. This presents a challenge to increasing everyday PA: people who lack awareness of their actions may not form credible plans for embedding PA into those actions. Raising awareness of routines may help people to understand their behaviour and thereby identify ways to achieve change.

‘Script Elicitation’ offers a novel method to modify routines performed non-consciously, on ‘autopilot’ [15]. Originally used as a purely descriptive tool to understand the processes through which consumers make purchasing decisions [16], Script Elicitation methods have more recently been adapted for application within health psychology as methods for both describing and modifying behavioural routines [15,17]. The Script Elicitation intervention method involves two stages. First, participants are supported to provide information regarding the structure and content of their routines (i.e., ‘scripts’), detailing the specific behaviours they perform, the typical sequencing of these behaviours, and potential cues to transitioning between behaviours within the sequence [15,16,17]. Second, the ‘script’ that is thereby elicited can subsequently be used as the basis to inform modifications to those routines [15]. To date, Script Elicitation has been administered via a one-to-one interview format, involving an interviewer collaborating with a participant to elicit a personalised, fine-grained action script [16]. Evidence suggests that Script Elicitation has the potential to change behaviour [15,17] via two mechanisms. First, reporting details of scripts requires participants to reflect on actions that they perform without thinking. While the notion of reflecting on actions enacted non-reflectively may appear paradoxical, people may possess procedural knowledge of their actions, such that they can be prompted to mentally rehearse and thereby describe the steps they follow in a sequence, yet lack declarative knowledge, such that they do not consciously think about the higher-level structure to these steps [18]. Reflection on specific actions raises participants’ awareness of those actions [15] and should thereby translate procedural knowledge of how an action is performed into declarative knowledge of the specific content and structure of the routine [18]. Second, by heightening awareness, Script Elicitation enables participants to plan changes to their habitually executed routines [13], by adding, removing or reorganising components. By supporting participants to develop a deeper understanding of their actions, the Script Elicitation procedure empowers them to modify those actions.

The one-to-one format in which Script Elicitation has been used to date is ill suited to a population-level public health intervention. However, if found to show promise when delivered one to one, future work would be warranted to develop a variant of Script Elicitation suitable for delivery at scale as a public health tool. Although the Script Elicitation intervention methodology is in its relative infancy, having not yet been trialled for effectiveness, two early-phase applications to date have shown promise. One study used the method to identify where best to insert flossing into oral health routines [17], and another showed that short sleepers found Script Elicitation helpful for understanding and removing unwanted elements of pre-bedtime routines [15]. Script Elicitation has, however, not yet been applied to PA. It is not certain that Script Elicitation would be suited to encouraging PA within everyday routines. Unlike Judah et al.’s intervention, for example, which focused on adding flossing into existing sequences of oral hygiene routines [17], commentators have called for PA to be incorporated into ostensibly unrelated patterns of everyday behaviour, such as commuting to work [9,19].

As a preliminary step in the development of a script-based PA promotion public health intervention, this small-scale, proof-of-concept study assessed the acceptability of using Script Elicitation to encourage PA uptake among office workers. We focused on office workers because they often fail to meet PA guidelines, spending around two-thirds of working hours sitting [20]. Among those who meet the guidelines, increased PA would likely achieve greater health benefits, possibly including offsetting the health harms of prolonged sitting [21]. Acceptability is an important dimension of intervention assessment [22], which can be assessed adequately and cost-effectively by using uncontrolled trial methods; if Script Elicitation were shown to be acceptable—i.e., perceived to be appropriate among potential recipients [23]—a more rigorous evaluation of its effectiveness would be warranted as the next step towards developing the intervention for public health purposes. Positive acceptability data from even a small, uncontrolled study would be sufficient to signify that it is worth pursuing script-based methods further. If found not to be acceptable, however, further reflection would be required regarding why participants responded unfavourably to script-based methods and whether it would be worthwhile to pursue the concept further.

## 2. Materials and Methods

### 2.1. Participants and Procedure

A pre–post, convergent parallel mixed-methods acceptability study design was used. Participants were office workers from a university and a university-affiliated student accommodation service in southeast England, recruited in May–June 2023. Eligibility criteria were aged 18 or over, working full-time in a predominantly desk-based job, and willing to increase their PA. Participants were recruited via an email sent to team managers at the university, who were asked to circulate information about the study to their team members. Study information included a URL or QR code linking to an online survey, through which participants self-reported their eligibility, demographics and job role, and email address and gave informed consent. Eligible respondents were contacted to arrange a baseline interview.

The baseline interview, which was run in person and one to one by one of four postgraduate psychology students (W.P., R.R.M., W.H.C.W., or X.N.), trained by B.G., included semi-structured questions about typical daily PA, followed by the Script Elicitation intervention procedure. Script Elicitation involved supporting participants to identify a period in their workday (before, during, or after work); generate a detailed description of actions undertaken within that period (i.e., a ‘script’); and co-design a version of this script involving more PA and plans to aid adherence to pursue each workday over the next week. Baseline interviews ran between 10 and 26 min (mean of 17 min; SD of 5).

One week later, each participant attended an in-person one-to-one follow-up semi-structured interview (same interviewer as at baseline), in which they reflected on their intervention experiences and responded orally to acceptability items. Follow-up interviews lasted 8–15 min (mean of 13 min; SD of 3).

On the day of each interview, participants were emailed a URL to an online survey to report their PA. Those who completed both interviews received an Amazon gift voucher worth GBP 5–25 (~USD 6–19), with values determined at random. All procedures were approved by the host institution’s ethics committee (FHMS 22-23 143 EGA).

Two team leaders circulated the email to staff, resulting in 11 participants (seven University-based and four student accommodation-based individuals; four males and seven females) aged 20–49 y (mean of 31 y; SD of 11). Participants were typically White British (n = 9; White European n = 1; Black/Black British: African n = 1), in administrative job roles (n = 7; managerial n = 3; academic n = 1). At baseline, they reported an average of 258 min/week of moderate or vigorous PA (SD of 255; moderate: 113 min/week, with SD of 106; vigorous: 145 min/week, with SD of 169) and 486 min/day of sitting (SD of 164), with five participants classified as ‘inactive’ and six as ‘active’ according to government guidelines (at least 150 min/week of moderate PA or 75 min/week of vigorous PA).

### 2.2. Script Elicitation Intervention

The intervention was run individually, privately, and face to face in a single session at the participant’s workplace. First, the interviewer worked with the participant to identify a specific period of time in their workday where they felt they could be more active (e.g., ‘during my lunch break’), drawing on participants’ responses to earlier interview questions about daily PA to guide them where necessary. Next, to elicit a script of the chosen period, participants were asked to identify the ‘start’ (e.g., ‘finish my meeting’) and ‘end’ points of that period (e.g., ‘log into computer’) and then describe in detail the sequence of actions undertaken between these points. Participants were then asked to cluster actions that they felt ‘go together’ (e.g., ‘leave the building’ and ‘fetch food’) into higher-order action categories (e.g., ‘get lunch’) and to identify potential cues that facilitate progression through these actions. To aid elicitation, the interviewer wrote all actions and cues on Post-It notes and arranged these on a table or wall visible to the participant. These processes continued until the interviewer and participant were satisfied that the current script was richly and comprehensively described.

Next, the interviewer and participant co-created an alternative version of the script to which they could feasibly adhere over the coming week. Participants were guided to introduce more PA into their routine by adding, removing, or reorganising actions. As participants generated ideas, researchers amended the current script by using differently coloured Post-It notes to highlight proposed modifications within the ‘alternative script’. If the participant struggled to identify PA-conducive activities to adopt, the interviewer provided a list of examples of vigorous- and moderate-intensity PA from a health service guidance document [3]. Participants could choose to add any PA activities to their alternative script, though interviewers emphasised the importance of planning only actions that participants believed they could feasibly and consistently enact within those routines. Where participants opted to insert new behaviours, or reorganise behaviours, they were asked to identify specific cues (e.g., location, time, and events) that could reliably precede the new or reorganised behaviours. The interviewer and participant co-created personalised if–then plans (i.e., ‘when I encounter X, I will do Y’), which were added to the alternative script as a reminder [24]. Participants were asked to try to adhere to their new script for at least seven days and, immediately after the interview, were emailed a flowchart depicting the alternative script, which they were advised to print and display in a prominent location.

### 2.3. Data Collection

All data are available at https://osf.io/sf4ej (accessed on 6 June 2024). Interview schedules and questionnaire items are provided as Appendix A.

*Quantitative data.* At baseline and at the one-week follow-up, *physical activity* was self-reported via the IPAQ [25], which assessed whether they engaged in moderate or vigorous PA, walking, and sitting and for typically how long, on each of the past seven days.

At follow-up, *perceived acceptability* was reported by using Theoretical Framework of Acceptability (TFA) items [23]. Six domains of acceptability were assessed, all rated on 1–5 scales (where 5 = most acceptable) and, unless stated otherwise, using a single item: affective attitude (two items), burden (reverse-coded for analysis), perceived effectiveness, intervention coherence, self-efficacy, and opportunity costs (reverse-coded).

*Qualitative data.* The baseline interview gathered qualitative data on current PA levels, typical periods of inactivity, and barriers to increasing PA. Follow-up interview data focused on experiences of Script Elicitation and behaviour change.

### 2.4. Acceptability Criteria and Analyses

Acceptability was examined against two criteria. If the intervention *reduces* PA, it cannot be deemed acceptable; thus, first, *mean PA levels should not decline, nor should sitting increase*. The study was neither powered nor intended to detect effects, so this was assessed via visual inspection of means, with paired-sample *t*-tests run for completeness. For exploratory purposes, we documented how many participants were physically active (i.e., achieving 150 min/week of moderate PA or 75 min/week vigorous weekly PA) at each timepoint. Second, *the mean of each acceptability score should be above the scale midpoint* (i.e., 3 on a 1–5 scale), indicating that participants tend to deem Script Elicitation acceptable. This was assessed by using descriptive statistics.

Interview transcripts, based on verbatim recordings, were analysed by using ‘codebook’ Thematic Analysis methods, underpinned by critical realist assumptions [26]. All authors independently inductively coded the same transcript for pertinent events, then met to agree on a coding structure of codes and preliminary clusters of codes (i.e., themes). Although we intended to derive the framework inductively, the data were found to fit a similar structure to Mohideen et al.’s study on the acceptability of Script Elicitation for sleep health [15]. The agreed coding framework, which was thus both inductively and deductively derived, was refined and applied by W.P. to the remaining transcripts. ‘In vivo’ codes were added to theme labels to demonstrate their grounding in the data. Senior author B.G. verified that the final analysis offered credible interpretations of the data.

## 3. Results

### 3.1. Description of Scripts

Participants most commonly chose to focus on scripts ‘during work’ (n = 5), with ‘before work’ and ‘after work’ each selected by three participants. Example current and alternative scripts are provided as Appendix A. Across all participants, the behaviour most commonly added to alternative scripts was going for a walk (n = 9). Walking the dog was selected by two participants, and going for a run and practising yoga were each added by one participant. Two preparatory behaviours (getting into active clothing and parking further from a destination) were each added by one participant. Behaviours commonly removed from or curtailed in scripts were browsing the internet (n = 3), scrolling on a phone (n = 3), and sitting (n = 3). One participant chose to curtail time spent lying in bed.

### 3.2. Quantitative Data: Acceptability

As Table 1 shows, time spent in all forms of PA increased, and sitting time decreased over the post-intervention week. Of the 11 participants, 6 were active at baseline and remained active at follow-up, 3 were inactive at baseline and became active at follow-up, and 3 were inactive at both baseline and follow-up; nobody who was active at baseline became inactive at follow-up. Additionally, all seven perceived acceptability scores were above the scale midpoint. Thus, both acceptability criteria were met.

### 3.3. Qualitative Data

Four themes were constructed. One related to perceptions of barriers to increasing PA at the pre-intervention baseline and three related to experiences of Script Elicitation, focusing on increased awareness, barriers to adherence, and perceived benefits.

*“There’s only so many hours in the day”: Pre-intervention perceived barriers to increasing PA.* Despite being motivated to increase their PA, participants felt their desk-based job limited opportunities for workplace PA (“I just sit at a desk all day and the most movement I do is walking to a meeting room and back”; Participant 7 [P7]). Work stressors, time pressure, and fatigue were seen to limit motivation outside of work (“[after work] I’m tired from work and I just want to relax and not think about [PA]”; P10). Perhaps consequently, participants felt that increasing their PA would be incompatible with their day-to-day priorities (“I try to do as much as I can, but there’s only so many hours in the day”; P8) and burdensome (“I have to make … an effort to be physically active”; P10).

*“It made me realise I have a routine”: Raised awareness of current routines.* For all participants, Script Elicitation highlighted that they enacted many of their day-to-day routines without conscious awareness (“*you’re just doing it rather than thinking about it*”; P10). Script Elicitation reportedly increased their awareness of the stability, consistency, and predictability of their routines (“*it made me realise that, although it can feel chaotic some mornings, I do actually have a [morning] routine*”; P4) and the typical lack of movement within such routines (“*it opened my eyes about how much I sit*”; P8). Some participants found the visual depiction of their routine especially insightful (“*seeing [my routine] all laid out simplifies it, and you think ‘I don’t really do as much [PA] as I think’*”; P4).

Many participants reported that Script Elicitation motivated and empowered them to plan feasible, PA-conducive changes to their routines:

“looking at what I’m doing already and breaking it down, you see that actually, I can add a little bit [more PA] there. And that’s [a] more realistic [PA plan] because it works around the stuff I’m doing most days already” (P2).

*“Life gets busy”: Barriers and facilitators to adhering to the alternative script.* The use of a flowchart depicting the new script was deemed helpful for reminding participants of their plan and consolidating their commitment to adhere to it:

“as soon as you put it down into [a] flow chart, it becomes real and it’s followed up, and … [it provides] the structure that [you] have to follow” (P6).

Yet, many participants reported various unforeseen and uncontrollable obstructions to adherence. For example, social events and work demands interfered with the stability of the overarching routines (“*things just came up with work that changed what I was doing and just meant I couldn’t do [the alternative script] regularly*”; P2). Perhaps consequently, one participant felt the flowchart “*simplified [my routine] too much*” (P5), by overlooking predictably unpredictable disruptions. For some participants, distractions prompted them to forget or momentarily deprioritise their alternative script (“*life gets busy … things happen*”; P3). Participants typically compensated for disruptions by executing their planned PA, or other forms of PA, within or outside of their target routine (“*technically I did make changes [to my behaviour], just not the ones I had planned*”; P4). Improvisation appeared to be facilitated by adoption of the principle of attempting to “*fit [more PA] in whenever I can*” (P2).

*“That’s an improvement for me”: Perceived benefits of Script Elicitation.* Many participants reported consistently adhering to their alternative routines as planned, so becoming more active. For some, their new routine was starting to feel more habitual nearer the end of the seven-day period (“*I didn’t feel like I needed the flow chart as I was just doing it automatically and getting on with it*”; P10). Gains in PA variously led to reports of feeling more energised, experiencing better sleep (“*I didn’t have a single nap this week, which is an improvement for me*”; P7), and improved mental health and wellbeing. Those who adopted PA within pre-work routines reported feeling more positive and productive at work.

## 4. Discussion

Achieving PA gains over the long term may require the adoption of PA into everyday routines, but many day-to-day routines are performed with minimal conscious thought. This small, proof-of-concept study, undertaken as a preliminary step in a broader public health intervention development project, examined the acceptability of a method designed to support people to understand their habitually executed routines and incorporate more PA into those routines. Our mixed-methods data showed that eliciting accounts of everyday routines (or ‘scripts’), together with supporting them to plan PA-conducive changes to those routines, was an acceptable intervention method among office workers. Self-reported PA did not decline between the baseline and the one-week follow-up, and participants rated the intervention favourably on seven indicators of acceptability. The interview data suggested that Script Elicitation drew participants’ attention to existing routines and aided development of PA-enhanced alternatives, in turn promoting PA and wellbeing. Our sample was small, and participants mostly already active. Nonetheless, findings suggest that script-based methods can potentially be adopted to modify PA. As a next step towards developing a script-based public health intervention suitable for delivery at scale, a more rigorous trial of Script Elicitation for PA is warranted.

This is the first study to use Script Elicitation intervention methods to encourage PA. Levels of PA did not lessen, nor did sitting increase, suggesting that at the very least, Script Elicitation caused no harm. Furthermore, although our study was not designed to detect effectiveness, participants posted considerable gains in walking and reductions in sitting time. Echoing findings from a recent acceptability study that used Script Elicitation to modify bedtime routines to enhance sleep [15], our participants reported that describing their routines in fine detail increased their conscious awareness of those routines and empowered them to plan how to modify those routines to incorporate more PA. These findings provide further evidence that office workers’ routines are often stable and predictable [20] and thus offer a useful vehicle for attempts to incorporate PA into everyday activities. They also demonstrate that many everyday routines proceed automatically [8], which poses a challenge to attempts to incorporate PA into those routines. Moreover, our results suggest that through guided self-reporting, people can gain new insights into PA-relevant routines. As Mohideen et al. [15] noted, Script Elicitation likely operates by helping people to convert procedural knowledge of *how* they perform their routines into declarative knowledge of *what* they do within those routines. Some participants reported unforeseen barriers to adhering to their intended new routines. This highlights a potentially key limitation of Script Elicitation. The method may perhaps be refined to encourage people to better anticipate potential obstacles to adherence when planning modifications to their routines [24]. However, for some participants at least, the intervention appeared to instil a broader mindset of seeking and seizing opportunities for PA in everyday activities. While further work is needed to assess effectiveness, our findings suggest Script Elicitation has the potential to help people to integrate incidental PA into their everyday routines.

Our findings establish a proof of concept: an approach that encourages people to reflect on their scripted, habitually executed routines and supports them to modify those routines was shown to be potentially suitable for adoption as a PA promotion strategy. Further work is needed before this intervention can be developed into a scalable public health intervention. For example, we delivered Script Elicitation one to one, which was resource- and time-intensive. This method may be well suited for incorporation into one-to-one support approaches such as motivational interviewing [27] but is unsuitable for delivery at the population level. If shown to be effective in future trials, work will be required to explore how to deliver Script Elicitation efficiently at scale. Such work might explore whether and how a variant of the Script Elicitation method might be self-administered, or employed by means of brief interviews undertaken via chatbot software, for example. Outside of the PA context, our study also provides further evidence to support the use of script-based methods as a means of modifying idiosyncratic health-relevant routines. While its effectiveness has not yet been definitively evaluated, Script Elicitation has now shown the potential to be applied to encourage flossing, sleep health, and in the present study, physical activity [15,17]. We encourage researchers to consider developing script-based approaches for public health initiatives in these and other health behaviour domains.

Study limitations must be acknowledged. A small sample and reliance on self-reporting prohibit robust conclusions. Similarly, in the absence of a control group, caution must be exercised in interpreting trends in PA over time following the Script Elicitation procedure, which may have occurred merely because of research participation [28]. Nonetheless, we obtained sufficient evidence of the potential for Script Elicitation to be acceptable to recipients as a PA promotion strategy, thus to progress to the next intervention development phase, at minimal cost. Also, our sample was highly active, reporting a daily average of 81 min moderate or vigorous PA. Although increasing PA among people who are already active may further enhance their health [5], greater public health benefit would be obtained by encouraging inactive people to adopt greater PA [19]. Also, our sample was recruited based on their motivation to increase their PA. This highlights a notable limitation of Script Elicitation, which presupposes positive prior motivation, so it is unlikely to be a useful standalone technique among those who do not want to be more active. If shown to be effective in future research, script-based methods would be best adopted as an adjunct to strategies that seek to motivate people to change their behaviour or as volitional support for people already motivated to increase their PA.

## 5. Conclusions

Our study suggests that supporting people to reflect on and make realistic modifications to their day-to-day, habitually executed routines offers an acceptable method for promoting PA among office workers. These findings represent an important step towards developing script-based methodology as a personally tailored public health tool for increasing PA. Progression to a more rigorous trial is justified as the next step in this development process.

## Figures and Tables

**Table 1 behavsci-14-00572-t001:** Physical activity and perceived acceptability scores (N = 11).

Measure	Baseline Mean (SD)	Follow-Up Mean (SD)	Difference	*p*-Value for Mean Difference
**Physical activity (minutes)**				
Vigorous PA (min/week)	145 (169)	188 (177)	+43	0.26
Moderate PA (min/week)	113 (106)	185 (140)	+72	0.12
Walking (min/week)	273 (145)	422 (202)	+149	0.02
*Total PA (min/week)*	*531 (350)*	*796 (328)*	*+265*	*0.01*
Sitting (min/day)	486 (164)	406 (171)	−80	0.02
**Perceived acceptability** (1–5; 5 = maximum acceptability)				
Affective attitude: liking		4.45 (0.69)		
Affective attitude: comfort		4.82 (0.41)		
Perceived burden *(reverse-coded)*		4.18 (0.75)		
Self-efficacy		4.73 (0.65)		
Perceived effectiveness		3.82 (1.17)		
Intervention coherence		4.27 (0.91)		
Opportunity costs *(reverse-coded)*		4.00 (1.41)		

## Data Availability

The data presented in this study are available at https://osf.io/sf4ej (accessed on 6 June 2024).

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
