# Peer review of "Can ‘Script Elicitation’ Methods Be Used to Promote Physical Activity? An Acceptability Study"

_behavsci, 2024, doi:10.3390/bs14070572_

Round 1

Reviewer 1 Report

Comments and Suggestions for Authors

It was a great pleasure that I reviewed the manuscript entitled “Can ‘script-elicitation’ methods be used to promote physical activity? An acceptability study” I think the paper presents some interesting findings. Here are several comments and suggestions that might help strengthen the paper.

1.     Although the Abstract section is well-written, I think a little more details for “Script Elicitation” should be described.

2.     Similarly, the Introduction section is well-written, I think a little more details for “Script Elicitation” should be described in the second paragraph of Page 2. I could tell what it is after reading the Methods section, but readers should be able to picture out at least at this section. Besides, I think a little bit about historical background of “Script Elicitation” is helpful, too.

3.     In the section of Materials and Methods, the authors described “Script Elicitation” as supporting participants to identify a period in their workday, generate a detailed description of actions undertaken within that period. It is my understanding that the period is PA, but exactly what kinds of PA can be included?

4.     In the same section, the authors also discussed the baseline interview involves co-design a version of this script involving more PA and plans to aid adherence etc. I think those types of things sound like intervention rather than just baseline assessment. Please clarify the purpose of the procedure.

5.     The last paragraph in the section of Participants and Procedure, the authors described the average PA scores assessed along with the baseline interviews via online surveys. Because there are two interviews (and surveys), are those scores averaged over two assessments? 

6.     I understand that the purpose of the study is not to detect effectiveness of “Script Elicitation”, but something the authors should clearly state as the reason is that there is no control group. I think it should be noted somewhere before the Results section.

Author Response

Comment 1: Although the Abstract section is well-written, I think a little more details for “Script Elicitation” should be described.

RESPONSE TO COMMENT 1: We now provide more detail of Script Elicitation (Lines 18-19), and have made edits to other aspects of the Abstract to ensure we meet the 200-word limit for the Abstract section.

Comment 2: Similarly, the Introduction section is well-written, I think a little more details for “Script Elicitation” should be described in the second paragraph of Page 2. I could tell what it is after reading the Methods section, but readers should be able to picture out at least at this section. Besides, I think a little bit about historical background of “Script Elicitation” is helpful, too.

RESPONSE TO COMMENT 2: We have expanded our description of the Script Elicitation method, including adding historical context (lines 83-92). Note too that we describe the historical context of Script Elicitation as a health behaviour change intervention in the following paragraph (lines 111-132).

Comment 3:  In the section of Materials and Methods, the authors described “Script Elicitation” as supporting participants to identify a period in their workday, generate a detailed description of actions undertaken within that period. It is my understanding that the period is PA, but exactly what kinds of PA can be included?

RESPONSE TO COMMENT 3: The ‘period’ we refer to here is a period of time, not a form of PA. We now clarify this on line 197. We also now clarify that participants could choose to add any form of PA into their ‘alternative script’, but that interviewers advised participants to think about the feasibility of any planned PA (lines 216-218).

Comment 4. In the same section, the authors also discussed the baseline interview involves co-design a version of this script involving more PA and plans to aid adherence etc. I think those types of things sound like intervention rather than just baseline assessment. Please clarify the purpose of the procedure.

RESPONSE TO COMMENT 4: We apologise for any miscommunication here. Script Elicitation as we have used it is an intervention procedure, which involves first assessing current behaviour, then undertaking an intervention in which participants are supported to plan changes to their behaviour. We have made several edits to clarify that Script Elicitation, as we have used it in this study, is an intervention procedure.

In the Abstract, we now describe Script Elicitation as ‘a novel intervention method’ (line 18).

In the Introduction, we state that Script Elicitation has originally been used for descriptive (i.e., non-intervention) purposes only, and then clarify that we are applying it as an intervention method (lines 83-87).

In the section referred to by the reviewer, we state that the baseline interview involved both semi-structured questions, and the ‘Script Elicitation intervention procedure’ (lines 165-168).

Comment 5. The last paragraph in the section of Participants and Procedure, the authors described the average PA scores assessed along with the baseline interviews via online surveys. Because there are two interviews (and surveys), are those scores averaged over two assessments? 

RESPONSE TO COMMENT 5: Thank you for pointing this out. We now clarify that these were baseline scores (line 189).

Comment 6. I understand that the purpose of the study is not to detect effectiveness of “Script Elicitation”, but something the authors should clearly state as the reason is that there is no control group. I think it should be noted somewhere before the Results section.

RESPONSE TO COMMENT 6: We now acknowledge the absence of a control group in the Introduction, by arguing for the adequacy of small, uncontrolled trials as low-cost methods for assessing acceptability (lines 143-149).

We also acknowledge the lack of a control group when discussing study limitations (lines 397-399).

Reviewer 2 Report

Comments and Suggestions for Authors

As an intervention study to promote physical activity, the method of applying script-elicitation is considered appropriate as a method to reflect on and modify an individual's routine behavior. However, I completely agree that there is a need to make up for the shortcomings mentioned by the researcher and hope that the methodology will develop.

Please supplement the following parts. (lines 64-68)

It was mentioned that people think about meaningful problems when performing routines. However, previous research has shown that people are more likely to think about problems unrelated to their own behavior. Researchers described the opposite concept. Habits can be considered unconscious actions, but routines require a definition as to whether they are conscious or unconscious actions.

The researcher's opinion expresses that routine is a conscious behavior that requires awareness of behavior. Please supplement this viewpoint.

Author Response

Comment 1. Please supplement the following parts. (lines 64-68)

It was mentioned that people think about meaningful problems when performing routines. However, previous research has shown that people are more likely to think about problems unrelated to their own behavior. Researchers described the opposite concept. Habits can be considered unconscious actions, but routines require a definition as to whether they are conscious or unconscious actions.

The researcher's opinion expresses that routine is a conscious behavior that requires awareness of behavior. Please supplement this viewpoint.

RESPONSE TO COMMENT 1. We have now clarified our position in several respects. Firstly, we provide a definition of a ‘routine’ as a “stable everyday pattern of behaviour” (lines 43-44). We recognise that this may seem non-committal with regards to whether these routines are conscious or non-conscious. However, leaving aside that there is no ‘correct’ or ‘incorrect’ definition of ‘routine’, it is beyond the remit (and word limit) of this paper to offer a discussion of whether the term ‘routine’ refers to a conscious or non-conscious pattern of behaviour.

Furthermore, this is a moot point because in the following paragraph, we shift our focus specifically to “routines [that] are done on ‘autopilot’, with little conscious awareness” (line 60). That is, regardless of whether the term ‘routine’ should mean conscious or non-conscious sequences, in this paper we are specifically referring to non-conscious routines. In the next paragraph, we also explicitly state that Script Elicitation is designed to “modify routines done non-consciously, on ‘autopilot’” (lines 82-83). Furthermore, we now explicitly state that “habitually executed routines, by nature, require little oversight [13], so people often think about *unrelated*, more meaningful matters when doing them [10]” (lines 76-77).

In recognition that it may seem odd to ask participants to reflect on behaviour that we believe they enact non-reflectively (i.e., non-consciously), we have added an explanatory sentence to the Introduction (lines 96-101).